# AN EMPIRICAL STUDY OF
# BINARY NEURAL NETWORKS' OPTIMISATION

**Milad Alizadeh, Javier Fernández-Marqués, Nicholas D. Lane & Yarin Gal**
Department of Computer Science
University of Oxford

## ABSTRACT

Neural networks with deterministic binary weights using the Straight-Through-Estimator (STE) have been shown to achieve state-of-the-art results, but their training process is not well-founded. This is due to the discrepancy between the evaluated function in the forward path, and the weight updates in the back-propagation, updates which do not correspond to gradients of the forward path. Efficient convergence and accuracy of binary models often rely on careful fine-tuning and various ad-hoc techniques. In this work, we empirically identify and study the effectiveness of the various ad-hoc techniques commonly used in the literature, providing best-practices for efficient training of binary models. We show that adapting learning rates using second-moment methods is crucial for the successful use of the STE, and that other optimisers can easily get stuck in local minima. We also find that many of the commonly employed tricks are only effective towards the end of the training, with these methods making early stages of the training considerably slower. Our analysis disambiguates necessary from unnecessary ad-hoc techniques for the training of binary neural networks, paving the way for future development of solid theoretical foundations for these. Our newly-found insights further lead to new procedures which make training of existing binary neural networks notably faster.

## 1 INTRODUCTION

There is great interest in expanding usage of Deep Neural Networks (DNNs) from running remotely in the cloud to performing local on-device inference on resource-constrained devices (Sze et al., 2017; Lane & Warden, 2018). Examples of such devices are mobile phones, wearables, IoT devices, and robots. This is motivated by privacy implications of sharing data and models with remote machines, and the appetite to apply DNNs in new environments and scenarios where cloud-inference is not viable. However, requirements of such devices are very demanding: there are stringent compute, storage, memory and bandwidth limitations; many applications need to work in real-time; many devices require long battery life for all-day or always-on use; and there is a thermal ceiling to consider when designing thin and light devices. On the other hand, the quest for more accurate DNNs has resulted in deeper, more compute-intensive models. This is particularly the case for CNNs. For instance, while the convolutional layers of AlexNet (Krizhevsky et al., 2012) make up only 4% of the model parameters, they are accountable for 91% of the computations at inference time (Louizos et al., 2017).

Compression and efficient implementation of DNNs are therefore more important than ever. There has been a spate of recent work proposing training and post-training schemes that aim to compress models without significant loss in their performance. Main examples of these techniques are pruning, weight sharing, low-rank approximation, knowledge distillation and perhaps most importantly quantisation to lower precisions (Han et al., 2017; Ullrich et al., 2017; Hinton et al., 2015). Quantisation is widely used in commercial deployments and its trade-offs and performance improvements for CNNs is well-studied in the literature (Krishnamoorthi, 2018). One appealing training-time quantisation scheme (Courbariaux et al., 2015) pushed it to the extreme, by representing each weight with a single bit, while maintaining respectable model accuracy. This paved the way for emergence Binary Neural Networks (BNNs). Courbariaux et al. (2016) and Rastegari et al. (2016) expanded BNNs by using the sign function as the non-linearity to achieve binary activations in addition to

binary parameters. With this approach, full-precision MAC operations in convolution layers can be replaced with cheap XNOR and POPCOUNT binary operations. This results in $58\times$ (Rastegari et al., 2016) improvement in compute-time in addition to the inherit $32\times$ saving in model size that comes from replacing 32-bit floating point parameters with binary ones.

However, as we will describe in Section 2, the common optimisation process used in BNNs is still not fully understood. Moreover, state-of-the-art binary models employ various modifications to conventional training settings in order to squeeze the best performance from the models. Some examples of these modifications are: applying constraints to weights and gradients, changing typical order of operations in a convolutional block, scaling learning rates based on Xavier (Glorot & Bengio, 2010) initialisation values, learning additional parameters for affine transformations of kernels, changing momentum hyper-parameters in Batch Normalisation (Ioffe & Szegedy, 2015) and the choice of optimiser, loss function, learning rate and number of training epochs. In the absence of rigorous mathematical understanding as of yet, it is imperative to empirically study the sensitivity of the optimisation process and the performance of BNNs to these settings and tweaks. Such an empirical understanding of the tools will greatly aid any development of solid mathematical foundations for the field. To that end, the main contributions of this work are as follows:

- We identify the essential techniques required for successful optimisation of binary models and show that end-to-end training of binary networks crucially relies on the optimiser taking advantage of second-moment gradient estimates.

- We show that most of the commonly used tricks in training binary models, such as gradient and weight clipping, are only required during the final stages of the training to achieve the best performance. Further, we demonstrate that these tricks lead to much slower convergence in the early stages of optimisation.

- We propose new procedures for training, making optimisation notably faster by delaying these tricks, or by training a full-precision model first and fine-tune it into a binary model.

- We provide our reference implementations and training-evaluation source code online [1].

## 2 BACKGROUND

Early quantised models were derived by quantisating full-precision weights of pre-trained models (Gong et al., 2014). This approach is widely used in real deployments and enjoys advantages such as the flexibility to apply different levels of quantisation based on the target model size, and not requiring knowledge of model internals. However, it suffers from a significant loss of accuracy. Hubara et al. (2017) showed that in order to maintain model performance, quantisation must be incorporated as part of the training process. This is done by either performing additional training steps to fine-tune a quantised model or by directly learning quantised parameters. This is essential for BNNs where binarising weights of pre-trained models result in significant loss in accuracy.

The first successful binarisation-aware training method was proposed in BinaryConnect by Courbariaux et al. (2015). In their work, the binary weights are not learned directly; instead, full-precision weights are maintained and *learned* during the training as proxies for the binary weights. These proxies are only required during training. During the forward path, binary weights are computed by applying $\mathrm{sign}$ function to their corresponding full-precision proxies. Since the $\mathrm{sign}$ function is not differentiable BinaryConnect employs Straight-Through-Estimator (STE) (Bengio et al., 2013) for back-propagating gradient estimates to full-precision proxies. The STE estimator simply passes the gradients along as if the non-differentiable operator was not present. In practice, BinaryConnect applied two additional restrictions on vanilla STE: (1) *Gradient clipping* stops gradient flow if the weight's magnitude is larger than 1. This effectively means gradients are computed with respect to $\mathrm{hard\,tanh}$ function. (2) *Weight clipping* is applied to weights *after* gradients have been applied to keep them within a range. To formalise this consider $w_r$ to be a full-precision proxy for binary weight $w_b$. During the forward path (and at the end of the training):

$$w_b = \mathrm{sign}(w_r)$$

---

[1] https://github.com/mi-lad/studying-binary-neural-networks

STE with gradient clipping provides an estimate for the gradient of this operation:

$$\frac{\partial w_b}{\partial w_r} = \mathbf{1}_{|r| \leq 1} \tag{1}$$

In a back-propagation context, we assume the gradient of the cost $C$ at the output ($\frac{\partial C}{\partial w_b}$) is available where in computing it the same STE estimator above has been used wherever required. Eq 1 enables us to estimate the gradient of the cost at the input ($\frac{\partial C}{\partial w_r}$) and update the proxies:

$$\frac{\partial C}{\partial w_r} = \frac{\partial C}{\partial w_b} \mathbf{1}_{|r| \leq 1}$$

The estimator passes gradients backwards unchanged when proxies are within the {-1,1} range and cancels the gradient flow when the proxy weight has got too positive or too negative. Figure 1 depicts how this works for a convolutional kernel in a CNN.

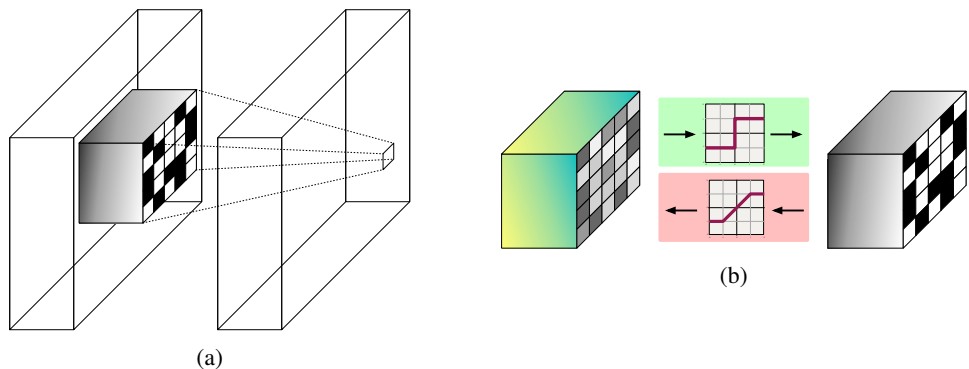

(a)

(b)

Figure 1: A convolutional kernel in a Binary Neural Network is binary (left) but its values are derived from a full-precision proxy learned using the STE estimator (right). At the end of the training, the proxy kernel is used for one last time to compute final binary values.

There have been several extensions to BinaryConnect's core idea of using STE estimator in binary models. BinaryConnect showed slightly better results when STE was used in stochastic binary neurons. BinaryNet (Courbariaux et al., 2016) used binary activations in addition to binary parameters (as described in Section 1) and made the convolution operation more efficient by using a custom GPU kernel. XNOR-Net and BWN (Rastegari et al., 2016) managed to scale up BNNs to achieve competitive results on the much bigger ImageNet (Deng et al., 2009) dataset by learning additional full-precision scale factors per-layer. DoReFa-Net (Zhou et al., 2016) used STE in the backpropagation path to quantise gradients and achieve faster training. TernaryNet (Zhu et al., 2016) quantised parameters to one and a half bits and represented weights using {-1,0,+1}. Having zero allows efficient hardware implementations when kernels are sparse. Lin et al. (2017) achieved state-of-the-art performance by learning a combination of very few binary kernels in each layer.

## 3 A SYSTEMATIC STUDY OF EXISTING METHODOLOGIES IN BNNS

There have been several non-empirical attempts to formalise STE and BNNs. Anderson & Berg (2018) took a high-dimensional geometric point-of-view to justify the existence of binary solutions irrespective of the optimisation process. Li et al. (2017) provided accuracy guarantees for training binary models under convexity assumptions. However, STE still has not been shown to find the solution of any particular loss function. In the meantime, binary models are achieving acceptable levels of accuracy in practice. Table 1 lists some of the recent binary architectures and their commonly-used training setup.

In this section, we provide an empirical analysis of the main approaches used in these models and help the researchers and practitioners navigate this space. We explore two classes of architectures

Table 1: Recent binary architectures and their training setup. The *Reorder* column refers to reordering of blocks in a convolutional layer to make sure pooling layer's input is full-precision. The *1^st Layer* column indicates whether the first layer of the network is binary or kept at full precision.

| Network | Optimiser | STE | Clipping | Reorder | 1^st Layer | Activation |
|---------|-----------|-----|----------|---------|-----------|------------|
| BinaryConnect | Adam | Yes | Weights & Gradients | Yes | Binary | 32-bits |
| BinaryNet | AdaMax | Yes | Weights & Gradients | Yes | FP | 1-bit |
| XNOR-Net | Adam | Yes | Gradients | Yes | FP | 1-bit |
| BWN | Momentum | Yes | Gradients | Yes | - | 32-bits |
| DoReFa-Net | Adam | Yes | Gradients | Yes | FP | ≥1-bit |
| ABC-Net | Momentum | Yes | Gradients | Yes | Binary | ≥1-bit |
| HBN | Adam | Yes | Gradients | Yes | FP | ≥1-bit |
| Bulat et al. (2017) | RMSprop | Yes | Gradients | Yes | FP | 1-bit |
| Cai et al. (2017) | Momentum | Yes | Gradients | Yes | FP | ≥2-bits |
| Xiang et al. (2017) | AdaMax | Yes | Gradients | - | FP | 1-bit |

in our study of binary networks: A CNN inspired by VGG-10 (Simonyan & Zisserman, 2015) on CIFAR-10 (Krizhevsky & Hinton, 2009) dataset and an MLP with three hidden layers with 2048 units and rectified linear units (ReLUs) for MNIST (LeCun, 1998) dataset. We make use of gradient and weight clipping and squared hinge loss unless stated otherwise. We use the last 10% of the training set for validation and report the best accuracy on the test set associated with the highest validation accuracy achieved during training. The results shown are the average of five runs. We have not used early stopping or finite time budget in any of the experiments.

The remainder of this section is organised as follows: We first show that the choice of optimiser matters considerably. We then show the impact of clipping gradients and weights followed by batch normalisation hyper-parameters on convergence speed and accuracy of BNNs. We finish by testing the effectiveness of some of the other commonly used tweaks used in training binary models.

## 3.1 IMPACT OF OPTIMISER

The majority of recent binary models use an adaptive optimiser in their implementations: BinaryConnect uses ADAM (Kingma & Ba, 2015) for CIFAR-10 and vanilla SGD for MNIST (although in their released source code they used ADAM for both datasets), DoReFa-Net and XNOR-Net use ADAM in their experiments and ABC-Net (Lin et al., 2017) uses SGD with momentum. In this section, we show that this is not accidental and investigate how the optimiser type and its associated hyper-parameters affects the viability of the STE estimator.

For experiments in this section, we looked at optimisers from four classes in order of increasing complexity: (1) history-free optimisers such as mini-batch SGD that do not take previous jumps or gradients into account, (2) momentum optimisers that maintain and use a running average of previous jumps such as Momentum and Nesterov (Sutskever et al., 2013), (3) Adaptive optimisers that adjust learning rate for each parameter separately such as AdaGrad (Duchi et al., 2011) and AdaDelta (Zeiler, 2012), and finally, (4) optimisers that combine elements from categories above such as ADAM which combines momentum with adaptive learning rate.

Table 2 summarises the best accuracies we achieved using different optimisers. We ran experiments for more epochs than typically required for the datasets (up to 500 epochs depending on the experiment). In each experiment, the relevant hyper-parameters were tuned for best results. We observed great variance in convergence speed and model performance as a result of optimiser choice that goes beyond differences seen when training non-binary models. Our first observation is that vanilla SGD generally fails in optimising binary models using STE. We note that reducing SGD's stochasticity (by increasing batch size) improves performance initially. However, it still fails to obtain the best possible accuracy. SGD momentum and Nesterov optimisers perform better than SGD when they are carefully fine-tuned. However, they perform significantly slower compared to optimsing non-binary models and have to be used for many more epochs than normally used for CIFAR-10 and MNIST datasets. Similar to SGD, increasing momentum rate improves training speed significantly but results in worse final model accuracy. In Appendix A we include results for the equivalent non-binary models that show the effect of batch size and momentum are far less substantial.

A possible hypothesis is that early stages of training binary models require more averaging for the optimiser to proceed in presence of binarisaton operation. On the other hand, in the late stages of the training, we rely on noisier sources to increase exploration power of the optimiser. This is reinforced by our observation that binary models are often trained long after the training or validation accuracy stop showing improvements. Reducing the learning rate in these epochs does not improve things either. Yet, the best validations are often found in these epochs. In other words, using early stopping for training binary models would terminate the training early on and would result in suboptimal accuracies.

Finally, adaptive optimisers, and specifically ADAM, consistently perform faster and able to achieve better accuracy levels. We experimented with different hyper-parameters in ADAM optimiser (see Figure 2c) and found the decay rate for the *second* moment estimate to play a significant role.

Table 2: Achievable test errors using different optimisers for binary MLP model trained on MNIST and binary CNN model train on CIFAR-10. The hyper-parameters of each optimiser were fine-tuned for best results.

|  | SGD | Momentum | Nesterov | AdaGrad | AdaDelta | RMSProp | ADAM |
|---|---|---|---|---|---|---|---|
| MNIST | 4.48% | 1.87% | 1.86% | 1.28% | 1.22% | 1.21% | **1.19%** |
| CIFAR-10 | 17.98% | 12.41% | 12.42% | 10.87% | 10.34 % | 10.33% | **10.30%** |

## 3.2 IMPACT OF GRADIENT AND WEIGHT CLIPPING

The STE variant used in BinaryConnect, XNOR-Net, and most other binary models, is different from vanilla STE introduced by Bengio et al. (2013). In these models, the STE stops gradient flow to proxies when the full-precision weights have grown beyond $\pm 1$. Additionally, BinaryConnect clips weights *after* gradient updates have been applied to keep weights within range. Our experiments (summarised in Table 3) show that this technique does indeed result in slight improvements in the accuracy of binary models. We observed 0.07% and 0.54% improvement for MNIST and CIFAR-10 datasets respectively. Clipping weights does generally help when it is combined with gradient clipping but is less effective on its own. In our experiments placing these additional constraints had negligible effects on speed of SGD or Momentum based optimisers. However, ADAM is sensitive to these constraints. We will revisit clipping in Section 4 where we study them again in terms of optimising convergence speed.

Table 3: Impact of gradient and/or weight clipping on the final test accuracy of BNNs.

| Clipping | None (Vanilla STE) | Weights | Gradients | Both |
|---|---|---|---|---|
| MNIST | 1.28% | 1.22% | **1.17%** | 1.18% |
| CIFAR-10 | 10.79% | 10.73% | 10.53% | **10.38%** |

## 3.3 IMPACT OF BATCH NORMALISATION

Batch normalisation (BN) uses mini-batch statistics during training but at inference-time the model is classifying a single data point. Therefore, each BN layer maintains a running average of mini-batch statistics to use during inference. The default momentum rate for this running average is usually large, e.g. 0.99. We noted that some binary models use smaller values for this hyper-parameter. Binary models are typically trained for more epochs than their non-binary counterpart and training is continued even when there is not a meaningful improvement in loss or accuracy. This is consistent with our earlier hypothesis in 3.1. Reducing the momentum rate in BN can help to cancel the effect of long training. The effect is small but consistent. Table 4 shows how different values of BN momentum results in different test accuracies. Krishnamoorthi (2018) also observed that Batch normalisation should be handled differently when training quantised models in order to achieve better performance.

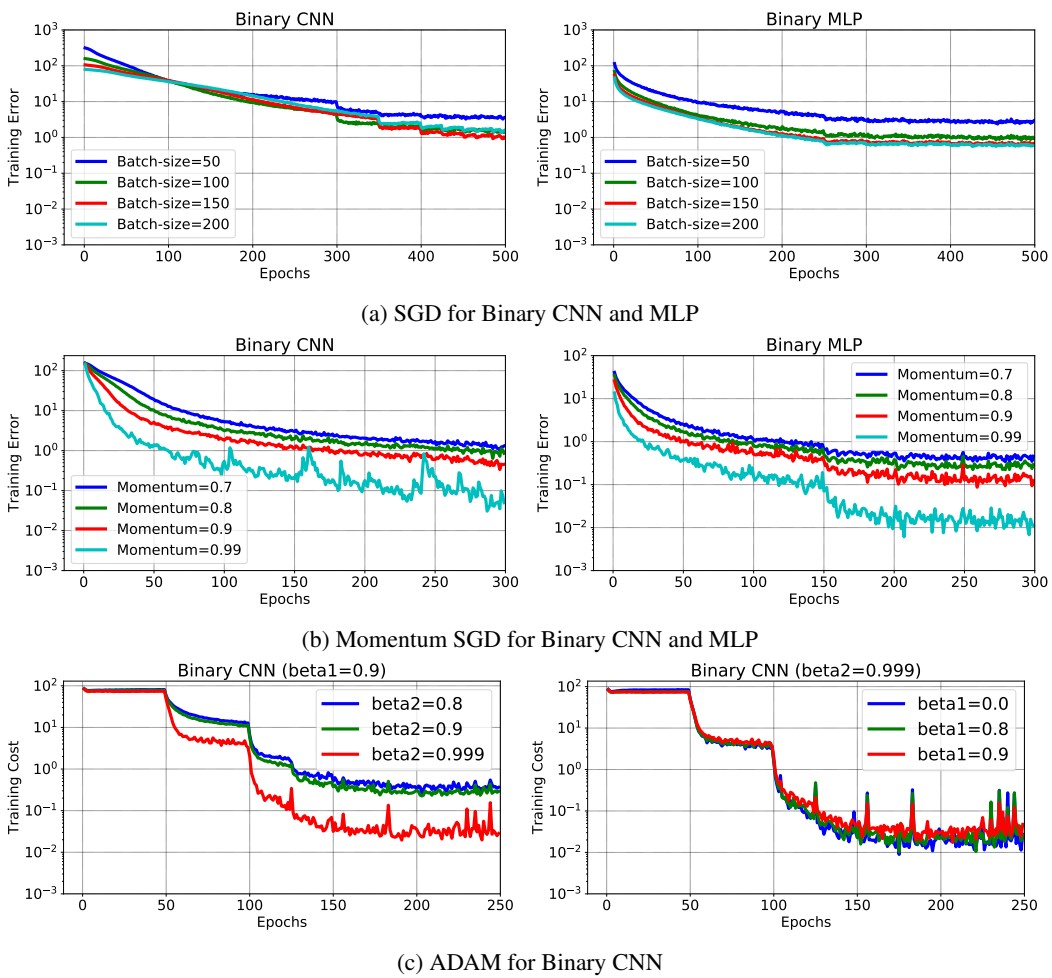

(a) SGD for Binary CNN and MLP

(b) Momentum SGD for Binary CNN and MLP

(c) ADAM for Binary CNN

Figure 2: Convergence speeds of two binary models trained with different optimisers. We found ADAM to be consistently faster in training BNNs compared to other optimisers. Figure (c) shows the effect of various momentum rates for ADAM's first and second moment estimates on the convergence of BNNs.

Table 4: Impact of momentum rate in Batch Normalisation's moving average on the final test accuracy of BNNs.

| Momentum | 0.8 | 0.85 | 0.9 | 0.99 |
|---|---|---|---|---|
| MNIST | 1.21% | **1.19%** | 1.22% | 1.23% |
| CIFAR-10 | **10.31%** | 10.35% | 10.53% | 10.61% |

## 3.4 IMPACT OF POOLING AND LEARNING RATE

**Reordering Pooling Block.** As can be seen in Table 1 all binary models change the placement of pooling operation within a convolutional layer. This change makes sense intuitively. For instance, applying MaxPooling to a binary vector results in a vector with almost all ones. We have seen two variants of block reordering and in both cases (see Figure 3), pooling is done immediately after the convolution operator where the vector is *not* binary. In our experiments, not making this change resulted in significant accuracy loss.

**Learning Rate Scaling using Xavier.** In BinaryConnect Courbariaux et al. (2015) propose scaling learning rates of each convolutional or fully connected layer by the inverse of Xavier initialisation's variance value. The same value is also used as the range in weight clipping after gradient update.



(a)          (b)          (c)

Figure 3: Changing the order of pooling operation within a convolutional block is necessary when training binary models.

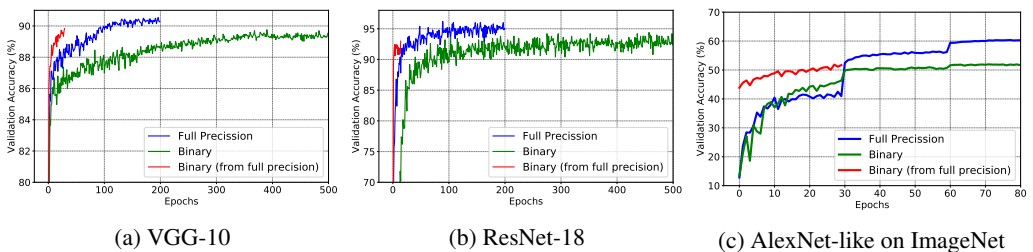

(a) VGG-10        (b) ResNet-18        (c) AlexNet-like on ImageNet

Figure 4: A binary model (red) is initialised from a full precision model (blue) and reaches top accuracy in a fraction of the epochs that would require to train a binary model (green) end-to-end.

They report noticeable accuracy improvements using these techniques. This modification is interesting because it suggests STE estimator requires an additional dimension. Applying this change effectively makes the slope of the line between $-1$ and $+1$ (see Figure 1b) directly proportional to the square root of (Fan-In + Fan-Out) of each layer. In our experiments, this approach helped when used with SGD but we did not see any impact when used with other optimisers. We were able to replicate accuracy levels reported by BinaryConnect without using this technique.

## 4   TRAINING BNNs FASTER: EMPIRICAL INSIGHTS PUT INTO PRACTICE

We continue in this section by applying a number of our empirical observations towards optimising BNNs in a faster and more efficient manner. We believe this demonstrates some of the practical implications of our results described earlier that are still to be explored.

In this case-study, we consider the well-known observation that training a binary model is often notably slower than its non-binary counter-part, the reasons for which are not well understood. One reason typically cited is that binarisation hinders the use of large learning rates – relative to those adopted in full precision networks. Our experiments show that counter to the conventional wisdom the STE on its own does not affect the training speed of BNNs considerably. The slowdown is mainly caused by the commonly applied gradient and weight clipping, as they keep parameters within the {-1,1} range at all times during training. Figure 5 shows how disabling one or both of these clipping schemes affects the training curve of a binary CNN. It can be seen that not clipping weights when learning rates are large can completely halt the optimisation (red curve in Figure 5). On the other hand, using vanilla STE brings the training speed back on par with the non-binary model. This is particularly true for ADAM.

However, this faster convergence comes at the price of a loss in accuracy (see Section 3.2). While weight and gradient clipping help achieve better accuracy, our hypothesis is that they are only required in the later stages of training where the noise added by clipping weights and gradients increases the exploration of the optimiser. We tested this hypothesis by training a binary model in two stages: (1) using vanilla STE in the first stage with higher learning rates and (2) turning clippings back on when the accuracy stops improving by reducing learning rate.

With vanilla STE the gradients are simply passed along to the full-precision proxies and the model is optimised as if the binarisation operations were not present. This, combined with results above, prompts the question of whether it is even necessary to apply binarisation from the very beginning of the training. While it might be conceptually attractive to train binary models end-to-end, we are

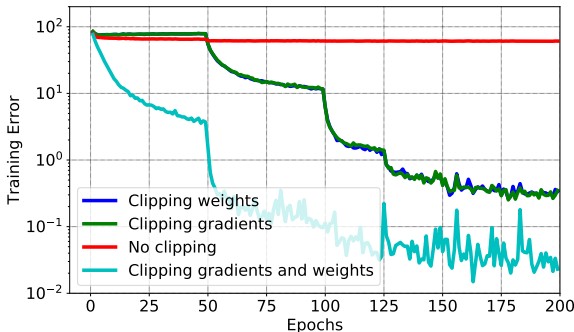

Figure 5: Impact of gradient and weight clipping on convergence speed of binary VGG-10 with large learning rates (0.1).

still learning full-precision structures during training. One can define and train an equivalent non-binary model where the binarisation operations are removed. This is useful because in many cases this model is already available. This pre-trained model can then be used to initialise the values of full-precision proxies in the binary model. The model can then be trained using STE and gradient clipping. Our experiments (see Figure 4) show that this works equally well in terms of accuracy but converges considerably faster for ResNet-18 (He et al., 2016) and VGG-10 architectures than if we had trained these binary models end-to-end. Mishra & Marr (2018) reported similar results.

There is a significant loss in accuracy when this model is binarised for the first time. This can be seen in the starting point of the *Binary (from full precision)* curves in Figure 4. This shows once again why we cannot simply binarise a pre-trained non-binary model and expect it to work well. However, we noted that the number of training steps required to recover the accuracy is very small. This result is encouraging because it turns the problem of learning binary models into a fine-tuning stage that can be applied to available pre-trained models.

It is important to note that while we can quickly get to the point where training and validation accuracies stagnate, there is a small gap between the achieved accuracy and the best possible one. This gap can only be filled by continuing training for many epochs. This difference is often consistent with the gap we observe between (a) the best test accuracy when training for many epochs and (b) the first epoch where validations accuracy stops improving. This reinforces our earlier hypothesis in Section 3.1 that suggests the last mile of model performance has little dependence on the STE's capability and mostly relies on a stochastic exploration of the parameter space.

Table 5: Training binary models using pre-trained full-precision models for CIFAR-10 (ResNet-18 and VGG-10) and ImageNet (AlexNet-like) datasets.

|  | Binarisation | Best Validation Accuracy | Test Accuracy |
|---|---|---|---|
| Binary ResNet-18 | end-to-end | 94.40% (in epoch 457) | 91.16% |
|  | from full-precision | 93.60% (in epoch **17**) | 91.18% |
| Binary VGG-10 | end-to-end | 89.76% (in epoch 391) | 89.18% |
|  | from full-precision | 90.16% (in epoch **24**) | 89.32% |
| Binary AlexNet-like | end-to-end | 51.98% (in epoch 88) | — |
|  | from full-precision | 51.85% (in epoch **30**) | — |

## 5 CONCLUSION AND BEST-PRACTICES

In this work, we study the landscape of binary neural networks and evaluate the impact of various techniques on the accuracy and convergence performance of binary models. We show that training binary models are harder and slower than the equivalent non-binary model. Our empirical study suggests that while the limit of STE's capability can be achieved easily, finding the best set of weights requires longer training. For efficient training of Binary models we recommend: (1) using

ADAM for optimising the objective, (2) not using early stopping, (3) splitting the training into two stages, (4) removing gradient and weight clipping in the first stage and (5) reducing the averaging rate in Batch Normalisation layers in the second stage.

ACKNOWLEDGMENTS

This work was supported by grants from EPSRC and ARM.

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

## A    IMPACT OF OPTIMISERS IN NON-BINARY MODELS

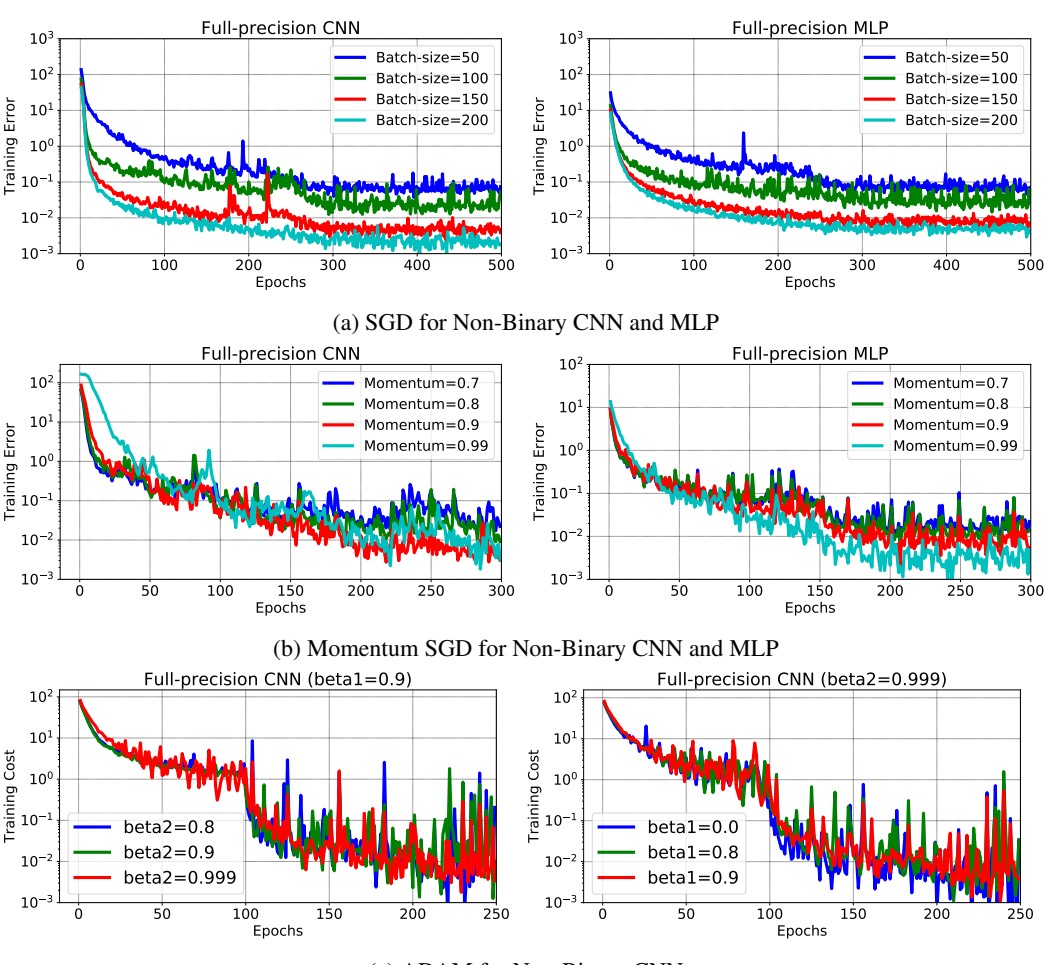

(a) SGD for Non-Binary CNN and MLP

(b) Momentum SGD for Non-Binary CNN and MLP

(c) ADAM for Non-Binary CNN

Convergence speeds of two full-precision models trained with different optimisers.

