# OpenReview forum: "An Empirical study of Binary Neural Networks' Optimisation"
_ICLR.cc/2019/Conference_

### Official Review · AnonReviewer1 · 2018-10-29
**The authors made several claims and provide suggestions on training binary networks. However, the experiments are somewhat not sufficient to support the proposed hypothesis.**

**Rating:** 4
**Confidence:** 4

**Review:**

The authors made several claims and provide suggestions on training binary networks, however, they are not proved or theoretically analyzed.  The empirical verification of the proposed hypothesis was viewed as weak as the only two datasets used are small datasets MNIST and CIFAR-10, and the used network architectures are also limited. Much more rigorous and thorough testing is required for an empirical paper which proposes new claims.

Take the first claim "end-to-end training of binary networks crucially relies on the optimiser taking advantage of second moment gradient estimates" as an example. As it is known that choice of optimizer is highly dependent on the specific dataset and network structure, it is not convincing to jump to this conclusion using the observations on two small datasets and limited network architectures.  E.g, many binarization papers use momentum for ImageNet dataset with residual networks. Does Adam also outperforms momentum in this case? Similarly, it is also hard for me to judge whether the other conclusions made about weight/gradient clipping, the momentum in batch normalization and learning rate, are correct or not.

Some minor issues are:
1. In Figure 4, different methods are not run to convergence, and the comparison may not be fair.
2. The second paragraph in section 4: "It can be seen that not clipping weights when learning rates are large can completely halt the optimisation (red curve in Figure 5)." However, in figure 5, the red curve is "Clipping gradients", which one is correct?
3. The authors propose a recipe for faster training of binary networks, is there experiments supporting that training networks with the proposed recipe is faster than the original counterpart?

---

> ### Author Response · Authors · 2018-11-19
> **Updated the paper to include some results on larger ImageNet dataset**
>
> We would like to thank the reviewer for the constructive comments.
>
> Our aim in this paper is to provide useful empirical observations and generate possible hypotheses that explain them, rather than to make new claims or theoretical analysis. It is true that we have provided some hypotheses about what might be going on, but at the end of the day, it is difficult to prove such new claims through empirical research. We did not aim to present conclusive observations for "X is necessary for Y" but rather give empirical support that "X seems to be tightly connected to Y", i.e. generating hypothesis to be validated in future theoretical research.
>
> > More datasets and architectures
> We do agree that in this line of work would benefit from more datasets and model architectures. We intended to repeat our experiments with larger datasets, but a hyperparameter search similar to what we have done for smaller standard datasets is computationally difficult on much larger datasets such as ImageNet dataset.
>
> However, we have now updated the paper to include results on ImageNet for Section 4 of the paper
>
> > Convergence in Figure 4
> This touches on the same points raised by another reviewer regarding early stopping. In our experiments, we tried to give the training phase in all experiments more than enough time to converge, but some optimizers (like vanilla SGD) simply fail in many scenarios to converge.
>
> >  In figure 5, the red curve is "Clipping gradients", which one is correct?
> Thank you for reporting this error. We have updated the paper to correct the order of items in the legend.
>
> > Baselines for training binary networks faster
> We believe these results are already included in Table 5 where "end-to-end" denotes the original counterpart experiments. Do let us know if you have something different in mind.

---

### Official Review · AnonReviewer2 · 2018-10-31
**Useful empirical study of existing methods**

**Rating:** 6
**Confidence:** 3

**Review:**

The paper systematically studies the training of binary neural networks, where binary in this case refers to single bit weight elements in the network. In particular, different existing training methods are tested and compared for training both MLPs and CNNs.

The main findings of the paper are:
- Using methods such as AdaGrad, AdaDelta, RMSProp and ADAM yields better performance than simpler momentum-based methods such as vanilla momentum and Nesterov momentum, which in turn are much better than vanilla SGD
- When training binary models, it is common to clip weights and/or gradients for the proxy weights in the network. In the paper it is however shown that these methods hinder using a fast learning rate in the beginning of training, while the methods are required in later stages of training in order to achieve good results
- Pre-training the model with full-precision training works well in speeding up training

For a practitioner, the paper presents a very useful reference for what methods work well when training binary networks. Although there are some proposals and hypotheses for reasons behind the results, I see the paper as a review paper of existing methods for training binary networks, showing experiments where the methods are tested using the same benchmark and training procedure in order to give a fair comparison.

As a practical guide, the paper therefore has clear value. What is lacking compared to typical ICLR papers is rigorously presenting new findings. The authors present a hypothesis for why different batch sizes are needed in the beginning compared to the end of training, but I found neither the justification nor the results very convincing with respect to the hypothesis. The way I see it, the actual novel proposals that are made in the paper are two two-stage training methods: one in which the tricks of weight and gradient clipping are only used towards the end of training, and one where the first stage of training is done using a full precision model. It is however quite well known that some training schemes with different stages can lead to improved performance: for instance with ADAM, even if it is an adaptive method, lowering the learning rate towards the end of training is often beneficial. It might therefore be fair to compare the methods to other multi-stage training methods. In addition, I could not find the training curves or final performance figures of the method where clipping is only activated towards the end of training.

To put it all together, the paper is clearly useful for the community as it provides a useful summary of the performance of different methods for training binary neural networks. In addition, it presents two two-stage training schemes that seem to make training even faster. What the paper lacks is rigorous theoretical justifications and clearly novel ideas.

Small comments:
- How are the training lengths decided for the different methods? If I am not mistaken, in Figure 2, it seems like the SGD and momentum methods have not yet converged when training is halted. Is there a budget for wall clock time or is early stopping used or something similar? Considering the nature of the paper, I would see this kind of decisions as important to report.
- In the abstract, you might want to refer to binary weights somewhere. Based on the abstract it is easy to mix the binary networks in this paper with stochastic binary networks that can also be trained using the STE estimator
- Are the differences in the performances in Table 3 statistically significant?

---

> ### Author Response · Authors · 2018-11-19
> **We need to know what works and what doesn't work before we can try to explain the things that work**
>
> We would like to thank the reviewer for the careful review and useful comments.
>
> We do agree on the reviewer's point that going forward, a rigorous understanding of these techniques is vital. There have been attempts to provide such theoretical justifications in the literature [1,2] but their scope has been limited. It was not our intention to provide such rigorous theoretical justifications in this paper, but we hope that our empirical work in identifying and understanding the effects of ad-hoc parameters and techniques could clarify things and form a precursor for such theoretical analysis.
>
> Regarding your specific comments:
>
> > Length of the experiments
> We have not used any budget limitation or early stopping in any of the experiments but this is a great point. In fact it makes a lot of sense to frame some our findings in the context of early stopping. In many cases we observed that once validation accuracy stops improving, there is often not a meaningful improvement in remaining training steps. We show how those extra steps allow squeezing a bit more accuracy from the model by training it for a very long time and relying on noise sources. Early stopping could be a good way to think about the actual capability of STE.
>
> We have updated the paper to make it clear where early stopping fits in our study. We have also updated the final “Best-Practices” section accordingly.
>
> >  Deterministic vs. stochastic binary weights
> Good point. The abstract has now been updated to make it clear that we are studying deterministic binary models and not the stochastic ones.
>
> >  Statistical significance of Table 3
> We have now updated the table in the paper to include the average results over 5 runs. Thank you for pushing us to do this.
>
> > Curves and figures for the two-stage clipping
> We will update the paper soon to include these figures.
>
> Finally, we have updated the paper to include results on ImageNet dataset for Section 4. Hopefully this will make our experiments more comprehensive.
>
> [1] Li, Hao, et al. "Training quantized nets: A deeper understanding." Advances in Neural Information Processing Systems. 2017.
> [2] Anderson, Alexander G., and Cory P. Berg. "The high-dimensional geometry of binary neural networks." arXiv preprint arXiv:1705.07199 (2017).

---

### Official Review · AnonReviewer3 · 2018-11-04
**Good review, relevant recommendations for a valuable research area**

**Rating:** 8
**Confidence:** 4

**Review:**

This is a good review paper covering techniques proposed across many of the well-known works in this area and doing an in-depth analysis of the value each of the techniques brings. Additionally, based on these studies the paper offers insights into the best algorithms and procedures to combine to achieving good results.

One recent whitepaper that has related work (not fully overlapping though), that may be worth looking by the authors is at https://arxiv.org/abs/1806.08342. It is fairly new and not very well-known so not surprising that the authors missed it.

Pros
- Well written paper with lots of in-depth experiments
- Does well at teasing out the impact of each of the techniques and gives some intuitive explanations of why they matter.
- Provides better insights into how to make training of binary neural networks faster.
- As the importance of low precision networks grows, this is a valuable paper in pushing the area of research forward.

Cons
- A review paper, which doesn't add much new to the existing suite of techniques. Note: This is true for most review papers.

---

> ### Author Response · Authors · 2018-11-18
> **Updated the paper to include the whitepaper**
>
> We would like to thank the reviewer for the constructive comments. We are glad the reviewer found our paper useful and well-written.  In agreement with the reviewer, we also believe that the characterization of compression techniques will become more and more important for the community as we go forward.
>
> Also, thank you for bringing the paper from the TensorFlow team [1] to our attention. This whitepaper is very relevant to the direction we have pursued in this paper. Interestingly, similar to our findings, it also identifies that Batch Normalisation should be handled differently when training quantized models to obtain better accuracies. It also looks at the consequences of using ReLU6 vs. ReLU. Another recent paper [2] recommended using PReLU to achieve better accuracy in Binary networks. It may be interesting to look more carefully at the choice of non-linearity function and its effects on the performance of quantized models.
>
> We have now uploaded a newer version of our paper in which we have embedded your comments. The paper now points to this whitepaper in the relevant parts (Sec 1 and Sec 3.3), and we think it has made it a lot better. We have also tried to make our empirical study more comprehensive by adding results on ImageNet dataset for Section 4.
>
> [1] Krishnamoorthi, Raghuraman. "Quantizing deep convolutional networks for efficient inference: A whitepaper." arXiv preprint arXiv:1806.08342 (2018).
> [2] Tang, Wei, Gang Hua, and Liang Wang. "How to train a compact binary neural network with high accuracy?." AAAI. 2017.

---

### Public Comment · (anonymous) · 2018-10-09
**Can you put up VGG-10 ？**

1.you said you use VGG-10 on cifar10 ，can you put up the  framework？
2. you know cifar10 is a small dataset, how about your expriments on ImageNet?

---

> ### Author Response · Authors · 2018-10-17
> **Response**
>
> Thank you very much for your comment. Both of your points are valid. We will definitely add details of the used architecture once we can make changes to the paper. We were planning to include results on ImageNet but unfortunately it did not make the deadline due to limited time and resources. It's difficult to train hundreds of model configurations on ImageNet dataset (as we've done for cifar-10) given its complexity. However, we will include some (but not comprehensive) results on ImageNet once the system allows us to update the paper.

---

### Meta-Review · Area_Chair1 · 2018-12-14
**Accept**

**Confidence:** 4
**Recommendation:** Accept (Poster)

**Metareview:**

The paper summarizes existing work on binary neural network optimization and performs an empirical study across a few datasets and neural network architectures. I agree with the reviewers that this is a valuable study and it can establish a benchmark to help practitioners develop better binary neural network optimization techniques.

PS: How about "An empirical study of binary neural network optimization" as the title?